# Polyvinyl Alcohol Polymer Functionalized Graphene Oxide Decorated with Gadolinium Oxide for Sequestration of Radionuclides from Aqueous Medium: Characterization, Mechanism, and Environmental Feasibility Studies

**DOI:** 10.3390/polym13213835

**Published:** 2021-11-06

**Authors:** Lakshmi Prasanna Lingamdinne, Janardhan Reddy Koduru, Yoon-Young Chang, Mu. Naushad, Jae-Kyu Yang

**Affiliations:** 1Department of Environmental Engineering, Kwangwoon University, Seoul 01897, Korea; swethasiri86@gmail.com (L.P.L.); yychang@kw.ac.kr (Y.-Y.C.); 2Department of Chemistry, College of Science, King Saud University, Riyadh 11451, Saudi Arabia; mnaushad@ksu.edu.sa

**Keywords:** PVA-GO-Gd composite, characterization, polyvinyl alcohol polymer, graphene oxide, gadolinium oxide, uranium, thorium, water treatment, adsorption mechanism

## Abstract

Uranium (U(VI)) and thorium (Th(IV)) ions produced by the nuclear and mining industries cause water pollution, thereby harming the environment and human health. In this study, gadolinium oxide-decorated polyvinyl alcohol-graphene oxide composite (PGO–Gd) was developed using a simple hydrothermal process to treat U(VI) and Th(IV) ions in water. The developed material was structurally characterized by highly advanced spectroscopy and microscopy techniques. The effects of pH, equilibration time and temperature on both radionuclides (U(VI) and Th(IV)) adsorption by PGO–Gd were examined. The PGO–Gd composite adsorbed both metal ions satisfactorily, with adsorption capacities of 427.50 and 455.0 mg g^−1^ at pH 4.0, respectively. The adsorption properties of both metal ions were found to be compatible with the Langmuir and pseudo–second-order kinetic models. Additionally, based on the thermodynamic characteristics, the adsorption was endothermic and spontaneous. Furthermore, the environmental viability of PGO–Gd and its application was demonstrated by studying its reusability in treating spiked surface water. PGO–Gd shows promise as an adsorbent in effectively removing both radionuclides from aqueous solutions.

## 1. Introduction

Uranium and thorium ions pollute groundwater and cause environmental and human health problems [1]. These radioactive wastes are produced as a result of nuclear fuel manufacturing and a range of industrial operations, including nuclear power stations, mining, nuclear arms, nuclear armament and laboratories dealing with radioactive elements. Radioactive ions may harm biological systems, and cause kidney damage, toxic hepatitis, damage to the histopathological system, skin corrosion and possibly cancer [2,3,4]. In addition, radionuclides (nuclear fuels) must be recovered from waste, which reduces the demand for nuclear power generation. Thus, a common and efficient method in treating radioactive-contaminated water is adsorption.

Graphene is a one-atom thick hexagonal sheet made up of sp^2^ hybridized carbon atoms tightly packed into a 2D honeycomb structure. It has great thermal conductivity, exceptional mechanical strength, and good electrical conductivity [5,6]. Graphene oxide (GO) is an oxidized form of graphene having a basal plane that is primarily changed with epoxide and hydroxyl groups, with carboxyl and carbonyl groups at the edges. Moreover, GO and GO-based materials are widely used in treating wastewater contaminated with heavy metals, radionuclides, and organic pollutants [7,8,9,10]. However, the real applicability of these materials in treating specific pollutants is limited because of their low functionality, high dispersibility, high aggregation, complex installation, and other physicochemical factors. However, surface modification of carbon materials with other materials leads to the enhancing of its stability feasibility for various sustainable applications [11,12]. For instance, the recent research on graphene-based polymer nanocomposites has opened a new avenue of study in the field of polymer nanocomposites [13]. Their properties are strongly connected to their nanostructures; homogeneous dispersion of nanocomposites and the exclusion of coalescence in the polymer matrices are critical for improving their properties [14]. Poly (vinyl alcohol) (PVA), a synthetic polymer derived from the parent polymer poly (vinyl acetate), has good chemical, physical, mechanical and thermal characteristics, as well as excellent film-forming abilities, non-toxicity and biodegradability [15,16,17]. PVA has been utilized in a variety of applications such as sealants, films, coatings, drug carriers, membranes and fuel cells, as well as in the commercial industries, medical and food industries [18,19,20,21]. PVA is a water-soluble polymer that may be combined with GO in a homogenous manner in water. As a result, the PVA/GO preparation procedure is reasonably easy and ecologically benign [22]. Furthermore, GO having oxygen-functional groups can affordably and efficiently induce full exfoliation and dispersion of GO homogeneously into the polymer matrices, enhancing the interfacial bond between GO layers and increasing surface sites while also being highly water dispersible. The capacity to accomplish complete exfoliation and uniform dispersion of GO in polymer matrices is required for the full use of GO layers in the applications of polymer nanocomposite [22,23]. Several studies have prepared PVA/GO, with a primary focus on environmental applications [24,25]. However, in these studies, the prepared PVA/GO were not used in removing specific radionuclides and exhibited a low removal capacity. However, this study mainly focused on enhancing the PVA/GO removal capacity targeting radionuclides, U(VI) and Th(IV), in addition to improving its stability, water dispersibility, surface functionality and feasibility for potential applications by functionalization with metal oxides.

Nowadays, due to their unique characteristics such as semiconductivity, paramagnetic nature, thermal stability and fluorescence, lanthanide series materials are increasingly used by several researchers in many applications, such as catalysis [26], adsorption [27] and energy storage [28]. Among the lanthanide elements, gadolinium oxide (Gd_2_O_3_) is regarded as an important compound in chemical and physical investigations; they are extremely effective supercapacitor catalysts [29], electrochemical sensors [30] and photodegradators [31] and adsorbers [32,33] due to their useful features and special properties, such as semi conductivity, thermal capacity, and low toxicity. These features of Gd_2_O_3_ enable its use as a good candidate for wastewater treatment.

In this work, we developed a simple, effective, and ecofriendly hydrothermal method to produce PVA/GO gadolinium oxide (PGO–Gd). The produced PGO–Gd composites were characterized by Fourier-transform infrared (FTIR) spectroscopy, thermal gravimetric analysis (TGA), scanning electron microscopy (SEM), X-ray diffraction (XRD), FT-Raman spectrum analysis, transmission electron microscopy (TEM), X-ray photoelectron spectroscopy (XPS) and Brunauer, Emmett and Teller (BET) analyses. Other essential parts of the adsorption mechanism, such as equilibrium, kinetics, pH, and thermodynamics, were also investigated, and the information gained from this research will enhance the adsorption process. This investigation clearly demonstrated the adsorption process involving radionuclides. This is the first report of a PVA-GO-modified adsorbent used for radioactive sequestration that we are aware of. Additionally, the prospective applicability and commercial viability of PGO–Gd were investigated by applying it to actual surface water and assessing its reusability. Moreover, the present material shows high removal capacity for radionuclides, and it can be re-used for up to four cycles without losing its removal efficacy of less than 60%.

## 2. Materials and Methods

### 2.1. PGO–Gd Preparation

GO was developed using a slightly modified Hummer’s technique [34]; 100 mg GO were dispersed in 100 mL water by ultrasonication and 100 mg PVA were dissolved in 50 mL water while stirring. The mixed GO and PVA solution was then heated to 120 °C for 5 h and cooled to 25 °C, and subsequently centrifuged (8000 rpm for 1 h). The collected sample was washed with water before drying at 60 °C, and named PGO.

PGO powder (100 mg) was dissolved in NaOH solution (6 g NaOH in 100 mL water) and ultrasonicated for 1 h to produce a clear mixture. Gd(NO_2_)_3_ (500 mg) was dissolved in 100 mL of water. The resulting gadolinium salt solution was dropped into the PGO solution while stirring. The reaction mixture was autoclaved at 120 °C. The final product was filtered and washed with water and ethanol before drying at 60 °C for 12 h. The product was labeled PGO–Gd. The method is schematically shown in Figure 1.

### 2.2. Removal Studies

The prepared PGO–Gd was used to sequestration of U(VI) and Th(IV) from water under batch studies. The impact of pH, contact duration, dosage, and metal concentration on U(VI) and Th(IV) adsorption was deliberated. For each batch study, 50 mL of a known concentration metal solution was placed in a 50 mL falcon tube and its pH was regulated to 4.0 before adding 0.1 g L^−1^ PGO–Gd. These tubes were subsequently agitated using a mechanical shaker at room temperature (25 ± 1 °C) for the prescribed equilibration period. The samples collected from the supernatant solution were separated from the PGO–Gd using 0.45 µm filters before being evaluated the concentration of U(VI) and Th(IV) using ICP-OES. At time intervals ranging from 5 to 400 min, the kinetics and effect of contact duration on metal ion sorption were investigated. The sorption isotherms and the impact of the metal concentration were studied by changing the initial metal ion concentration from 5 to 100 mg L^−1^. The pH impact was examined by changing the solution pH from 2.0 to 8.0 with a dilute HCl and NaOH solutions. All batch studies were conducted in duplicate, and the results shown are the averages of two measurements. The adsorption capacity and % removal can be calculated based on Equations (1) and (2):(1)$$\mathrm{Adsorption}\;\mathrm{capacity},\;q_{e}=\left(C_{o}-C_{e}\right)\times v/m\;$$
(2)$$\mathrm{Adsorption}\;\mathrm{removal}\;\%=\left(\frac{C_{o}-C_{e}}{C_{o}}\right)\times 100\;$$
here showing the metal sorption capacity (*q_e_*, mg g^−1^), metal initial and equilibrium concentrations (*Co* and *Ce*, mg L^−1^), volume of solution (*v*, L) and adsorbent mass (*m*, g).

### 2.3. PGO–Gd Regeneration and Application to Surface Water Samples

The PGO–Gd stability and re-usable feasibility were evaluated through approximately five cycles of reuse. The metal sorption capacity (qe, mg g^−1^), as shown in Equation (1), was evaluated batch-wise at pH 4.0, 298 K, and 0.1 g L^−1^ adsorbent to assess the stability of the adsorbent across the cycles.

Surface water was obtained from Seongu-ri, Onjeong-myeon, Uljin-gun and Gyeongsangbuk-do (Republic of Korea) was used for evaluation of prepared material feasible applicability in real system. The specifics are provided in the Appendix A). The pH, cations (Ca^2+^ and Na^+^) and anions (PO_4_^3−^, SO_4_^2−^, NO_3_^−^, Cl^−^ and HCO_3_^−^) of the surface water were all measured.

## 3. Results and Discussions

### 3.1. PGO–Gd Structural Chracterization

XPS analysis was used to determine the surface electronic states and compositions of GO and PGO–Gd (Figure 2a). The examined XPS peaks clearly shows C, O and Gd element signal, which are compatible with the EDX findings (Appendix A). In Figure 2a, the peak of C 1s signal is observed at 283.49 eV, and the signal at 529.37 eV is attributed to O^2−^, which corresponds to GO [35]. The Gd 4d XPS plot indicates that the signal at 141.4 eV is attributed with Gd^3+^ [36,37]. Hence, the XPS analysis confirmed the formation of PGO–Gd.

The XRD patterns of GO and the PGO–Gd composite are given in Figure 2b. GO shows strong peaks at 2θ ≈ 11.3°, which corresponds to the (002) planes. The peaks labeled by triangles can be directly indexed to the cubic phase structure of Gd_2_O_3_ (JCPDS no. 12-0797). This indicates the successful integration of Gd_2_O_3_ into PGO [38]. The TGA approach was used to measure the amount of oxygenated functionals on the surface of GO and PGO–Gd by oxidative decomposition from 25 °C to 1200 °C at a rate of 10 °C min^−1^, as illustrated in Appendix A. The resultant TGA plots shows the three phases of the weight decrease in the TGA curves of GO. First, a little mass reduction at below 100 °C can be due to adsorbed water molecules evaporating. Second, a substantial mass loss from 100 to 350 °C can be ascribed to the elimination of labile oxygenated functional groups of GO including epoxy, carboxyl, and hydroxyl vapors. Finally, the decomposition of GO caused a modest loss of weight from 350 to 1200 °C. Furthermore, when the GO and PGO–Gd of TGA curves were compared, the weight loss of GO and PGO–Gd at 0–350 °C were approximately 37% and 16%, respectively. Hence, both GO and PGO–Gd are presumed to contain an abundance of oxygenated surface functional groups, and that PGO–Gd, some of oxygen-containing groups replaced with Gd_2_O_3_ nanoparticles. At 1200 °C, weight loss for GO was ~67%, and PGO–Gd was observed 26%, which was related to the formation of cubic Gd_2_O_3_ by the phase transition from hexagonal Gd(OH)_3_ of PGO–Gd [38]. These results suggest a higher stability and surface functionality of PGO–Gd than that of GO.

Appendix A indicates the FTIR spectra of GO and PGO–Gd. The stretching vibrations of –OH (3339 cm^−1^), C=C (1634 cm^−1^), C–O (1217 cm^−1^) and C–O–C (1054 cm^−1^) were discovered by an FTIR scan of GO. The PGO–Gd spectra displayed stretching vibrations at 3617 cm^−1^ (–OH), 1515 cm^−1^ (C=C), 1380 cm^−1^ (C–OH), and 1217 cm^−1^ (C–O–H) and Gd_2_O_3_ spectra observed Gd–O stretching (514 cm^−1^), and this is consistent with prior results [36]. Furthermore, the strong stretching peak at 710 cm^−1^ indicates Gd(OH)_3_ [39]. Raman investigation confirmed the existence of substantial GO in the PGO–Gd composite. The Raman spectra of PGO–Gd were compared with those of GO for this purpose, as illustrated in Appendix A. In general, GO has two major signals in the Raman spectra: the G and D bands at 1598 and 1353 cm^−1^, respectively. Chemically decorated Gd_2_O_3_ on the GO lattice surface exhibited a peak shift and appeared in the D-band at 1347 cm^−1^, and an expanded G–band at 1590 cm^−1^. A minor difference in the I(D)/I(G) (are the peak intensity ratio of the D– and G–bond) of PGO–Gd (0.93) comparing with that of GO (0.96) suggested that gadolinium oxide particle surface modification changed the in-plane sp^2^ graphitic domains of GO. These results are comparable to those obtained with GO coupled with metal oxide nanoparticles and biomolecules [40].

The SEM picture of PGO–Gd in Figure 3a,b indicates that the severely aggregated Gd_2_O_3_ nanoparticles are well dispersed over the PGO surface. A strong electrostatic attraction between the PGO and the Gd_2_O_3_ nanoparticles may help to maintain the composite form. This might be due to the PGO’s active surface area and oxygen moieties being drawn to the Gd_2_O_3_ nanoparticles. The TEM image of PGO–Gd is presented in Figure 3c,d, wherein the aggregated Gd_2_O_3_ particles are distributed on the corrugated thin sheet-like membranous layer of the PGO surface because of the paramagnetic nature of gadolinium. Further the TEM results suggest rod chape cubic crystalline with an average diameter of 30–40 nm and a nearly uniform distribution with less aggregation. As a consequence, the TEM results are compatible with the SEM findings and showed the composite’s formation at nano level crystals with rod shape. BET analysis provides further information on the surface area and pore structure, wherein the surface area of PGO–Gd was 85.30 m^2^ g^−1^, and the pore diameter and pore volume were 21.37 nm and 0.18 cm^3^ g^−1^, respectively. The BET results suggest the prepared crystalline PGO–Gd possesses a mesoporous surface morphology.

### 3.2. pH Impact on Adsorption

As given in Figure 4a, the pH of the aqueous solution considerably affected both radionuclides sorption on to PGO–Gd surface. The adsorption percentages of U(VI) and Th(IV) progressively rose in the pH range of 2.0–4.0, reaching 99% at pH 4.0. Upon further increasing the pH from 4.0–8.0, the removal percentage remained constant at high values. The increased sequestration of both radionuclides by PGO–Gd with solution pH increases may be ascribed to the PGO–Gd surface charges and dissociation of surface functional groups as well as the dispersion of U(VI) and Th(IV) species in solution. At pH < pH pzc (5.75), PGO–Gd surface pronated (i.e., ≡SOH + H^+^ → ≡SOH_2_^+^) and produce positive surface charge. Thus, the small adsorptive efficiency of PGO–Gd is ascribed to the electrostatic repulsion between U(VI) and Th(IV) at pH 0.5–3.5, and UO_2_OH^+^, (UO_2_)_2_(OH)_2_^2+^, (UO_2_)_3_(OH)_5_^+^, (UO_2_)_4_(OH)_7_^+^, Th(OH)_3_^+^, and Th_2_(OH)_2_^6+^ at pH > 4.0 [4,9], and the positively charged edge functional groups (SOH_2_^+^) on the surface of PGO–Gd. However, due to the deprotonation process (i.e., ≡SOH → ≡SO^−^ + H^+^) the PGO–Gd surface becomes negatively charged at pH > pHpzc, which enhances the electrostatic attraction between positive charge ions (U(VI) and Th(IV): UO_2_OH^+^, (UO_2_)_2_(OH)_2_^2+^, (UO_2_)_3_(OH)_5_^+^, (UO_2_)_4_(OH)_7_^+^, Th(OH)^3+^, and Th_2_(OH)_2_^6+^ at pH > 4.0) and negatively charged SO^−^ groups on PGO–Gd, increasing the proportion of both radionuclides adsorption [4]. This result is further supported by examining the pH of the treated solution; under acidic conditions, protonation reaction on the surfaces of PGO–Gd slightly increased the solution pH values, whereas under alkaline conditions, deprotonation reaction on PGO–Gd surface decreased the solution pH values. Furthermore, at high pH, the dissociation of surface functional groups of PGO–Gd leads to the formation of more negatively charged surface sites that facilitate the binding of U(VI) and Th(IV) ions. The features of U(VI) and Th(IV) species that predominate at a given solution pH may have a significant impact on PGO–Gd removal effectiveness for U(VI) and Th(IV). The relative distribution of U(VI) and Th(IV) species according to hydrolysis constants from prior literature can fairly explain the sorption behavior of U(VI) and Th(IV) [9]. From the results of the overall pH effect, a pH of 4.0 was chosen as the optimum condition in the present study.

### 3.3. Kinetics of Adsorption

The adsorption features of the PGO–Gd adsorbent for radionuclide ions was evaluated by varying the adsorption rate of U(VI) and Th(IV) with time. Figure 4b presents the results of the kinetic adsorption of U(VI) and Th(IV) by PGO–Gd. The adsorption rate of U(VI) ions on PGO–Gd increased rapidly during the initial stages of adsorption, reaching approximately 94% within 60 min and attained equilibrium within 120 min. Th(IV) adsorptive removal by PGO–Gd was rapid, and 99% removal occurred within 5 min. The kinetics for the adsorption of U(VI) and Th(IV) ions were determined using the following pseudo–first-order (PFO) (Equation (3)) and pseudo–second-order (PSO) (Equation (4)) models:(3)$$\mathrm{PFO}\;\mathrm{kinetic}\;\mathrm{equation}:\;q_{e}=q_{e}\left(1-10^{-k_{1}\cdot t/2.303}\right)$$
(4)$$\mathrm{PSO}\;\mathrm{kinetic}\;\mathrm{equation}:\;q_{e}=(k_{2}\;t\;q_{e\;}^{2})/\left(1+k_{2}\;t\;q_{e}\right)$$

Table 1 lists the adsorbed quantities derived from the kinetic models (*q_e_,_model_*), experimental (*q_e,exp._*), and other kinetic parameters. The significance of the correlation coefficients (R^2^) was close to 1, and the closeness of the *q_e_,_model_* and *q_e,exp_*. values indicate that the adsorption kinetic data of U(VI) were well explained by PSO model. This results suggest that the U(VI) sorption was rate limits PSO kinetics, whereas, Th(IV) was rapid and was time-independent and have 100% removal within 5 min.

### 3.4. Equilibrium Isotherms

The maximum adsorption capacity of PGO–Gd for U(VI) and Th(IV) was determined by changing metal ion starting concentrations from 10 to 100 mg L^−1^ while keeping all other parameters constant (0.1 g L^−1^ adsorbents, 50 mL U (VI) and Th(IV) solution, PGO–Gd pH at 4.0 and temperature at 25 °C). The equilibrium isotherm data of both metal ions were plotted in Figure 4c, and the resultant data were simulated using the Langmuir, Freundlich and Temkin isotherm models (Appendix A) to elucidate the adsorption process. The Langmuir, Freundlich and Temkin isotherm models are expressed in Equation (5), (6) and (7), respectively.
(5)Ceqe=1qmKL+Ceqm
(6)$$logq_{e}=\log K_{F}+n^{-1}\;logC_{e}$$
(7)$$q_{e}=B\log K_{T}+B\log C_{e}$$

Table 2 lists the Langmuir, Freundlich and Temkin isotherm parameters determined through the fitting procedure. The Langmuir equation, with a higher correlation coefficient (R^2^) of 0.998, fits the experimental data better than the Freundlich and Temkin models, signifying that the adsorption process was the monolayer on the homogeneous surface of PGO–Gd for both metal ions. The resultant maximal removal capacity of PGO–Gd was found to be 427.50 and 455.0 mg g^−1^ for U(VI) and Th(IV), respectively. This high adsorption capacity indicates that PGO–Gd is an effective sorbent in removing U(VI) and Th(IV). The results suggest that the surface functional groups are presumed to play significant roles in the efficient exclusion of U(VI) and Th(IV). However, the higher adsorption capacity of Th(VI) than U(VI) is may be due to a difference in the ionic radius, in the pKas and in the speciation’s. With increasing ionic radius, the steric crowding on the adsorption surface will also increase; thus, a saturation limit of adsorption is rapidly attained.

The effect of temperature on U(VI) and Th(IV) adsorption by PGO–Gd was determined by conducting adsorption experiments at 298, 313, and 333 K. With increasing temperature, the sorption efficiency of both metal ions increased, suggesting that the adsorption process is endothermic. The slope and intercept of the plot of *lnKc* vs. *T* (Figure 4d) revealed the standard enthalpy change (ΔH^0^) and entropy change (ΔS^0^), respectively, and the Van’t Hoff equation revealed the Gibbs free energy change (ΔG^0^) and was listed in Appendix A. The resultant negative ΔG^0^, and positive ΔH^0^ and ΔS^0^ indicate the feasibility of adsorption. However, the fact that adsorption capacities increase with temperature suggests that chemical interactions between the metal ions and PGO–Gd are the primary determinant of both radionuclide adsorptive removal [4,41].

### 3.5. Comparison of Adsorption Capacities and Cost

Table 3 compares the Langmuir q_max_ of PGO–Gd for U(VI) (427.50 mg g^−1^) and Th(IV) (455.0 mg g^−1^) removal with those of other possible adsorbents described in the literature. The q_max_ of PGO–Gd for both metal ions were equivalent to and somewhat higher than that of numerous other adsorbents described in the literature, according to Table 3. In addition, the adsorbent PGO–Gd was produced in this work using a simple hydrothermal technique and was recyclable or reusable for more than four cycles (Appendix A) without reducing its original efficacy and stability. As a result, when compared with other adsorbents reported in the literature, the PGO–Gd cost is expected to be low. Furthermore, the cost, stability, and adaptability of the adsorbent employed determine the effectiveness of any adsorption method in water treatment. As a result, the material developed in this work has the ability to effectively treat water polluted with U(VI) and Th(IV).

### 3.6. Adsorption Mechanism

The use of XPS characterizations aids in understanding the roles of different surface functional groups of PGO–Gd in the adsorption of U(VI) and Th(IV). As shown in Figure 5a–c, the characteristic peaks of the XPS full survey, C 1s and O 1s, respectively, can be observed in PGO–Gd before and after adsorption. After U(VI) adsorption, the double characteristic peaks of U 4f and Th 4f appeared in the sample spectra, suggesting effective immobilization of U(VI) and Th(IV) on the adsorbent. The high resolution C1s spectra (Figure 5b) was well resolved by three peaks located at 283.31 eV (C=C or C–C), 284.96 eV (C–OH), and 287.35 eV (C=O). The high resolution O1s spectra (Figure 5c) was fully resolved into three separate component peaks located at 531.49 (C=O) and 529.87(O=C–OH). Adsorption of metal ions altered the C1s (C=C, C–OH, and C=O) and O1s (C=O and O=C–OH) peak positions, suggesting that U(VI) was predominantly adsorbed by interacting with PGO–Gd surface functional groups such as C=C, C–OH, and C=O. Moreover, Th(IV) was predominantly adsorbed by interacting with PGO–Gd surface functional groups, such as C=C, C–OH, O=C–OH, and C=O, which probably contributes to the rapid adsorption of Th(IV). Furthermore, the adsorption studies, such as pH, kinetics, and isotherms, suggest that both metal ions were adsorbed via electrostatic interactions and surface chemical complexation. However, chemical complexation predominantly influenced the sorption process for both U(VI) and Th(IV). The plausible sorption mechanism is illustrated in Figure 6.

### 3.7. Studying Environmental Relevance by Treating Metal Spiked Real Surface Water

Additional studies were performed for evaluating the adsorption stability of PGO–Gd via kinetic experiments to investigate the in-situ applicability for surface water remediation. The stability experiments were performed by adding 0.005 g of adsorbent to 50 mL of 10 mg L^−1^ U(VI) and Th(IV)-spiked groundwater, and 5 mL of each supernatant was collected at varying time intervals, filtered, and the residual concentration was estimated by ICP-OES analysis. Compared with U(VI), Th(IV) removal reached the EPA standard level within 5 min (Figure 7), whereas the U(VI) residual concentration in treated water did not reach the EPA standards, indicating the influence of other associate ions, causing lower U(VI) adsorption than that of Th(IV). Moreover, Th(IV) adsorption was unaffected by U(VI) and vice versa, when the groundwater was spiked with mixed metals (Th(IV) and U(VI)) (each contaminated at 10 mg L^−1^). These results suggest the selective sorption of both metal ions under the present experimental conditions.

## 4. Conclusions

In this work, we focused on the facile preparation of PGO–Gd, and studied their sorption mechanism for U(VI) and Th(IV) in ground water. Prepared PGO–Gd samples were well characterized by XRD, XPS, FT-IR, Raman and SEM-EDX, which revealed their structure and purity. The sorption characteristics of both metal ions from aqueous solutions were observed under numerous experimental conditions that included pH, kinetic dosage, isotherms, and temperature techniques.

The results are concluded in the following remarks.

(1)Th(IV) and U(VI) absorption was comparably high in PGO–Gd, although Th(IV) adsorption was somewhat greater than U(VI) (VI).(2)The effect of varied pH values on metal intake revealed that increasing the pH enhanced metal ion absorption by PGO–Gd, reaching a maximum at pH 4.0. This finding also suggests that the surface charge and metal ion species have a significant impact on the adsorbent’s sorption capacity.(3)The *q_max_* of PGO–Gd for U(VI) and Th(IV) was comparable and higher than those of the other absorbents.(4)The adsorption process is endothermic and thermodynamically favorable.(5)The PSO kinetic model and the Langmuir isotherm accurately explain the sequestration of U(VI) and Th(IV) by PGO–Gd, suggesting that the rate-limiting monolayer sorption process happened on the PGO–Gd homogeneous surface.(6)The characterization and adsorption studies concluded that the ions were adsorbed predominantly by surface complexation along with electrostatic interactions through adsorbent surface functionality.(7)The adsorbent can be reused up to four times without losing its original efficacy or stability. Hence, the use of PGO–Gd to remove radioactive waste from surface water is strongly recommended in this study.

## Figures and Tables

**Figure 1 polymers-13-03835-f001:**
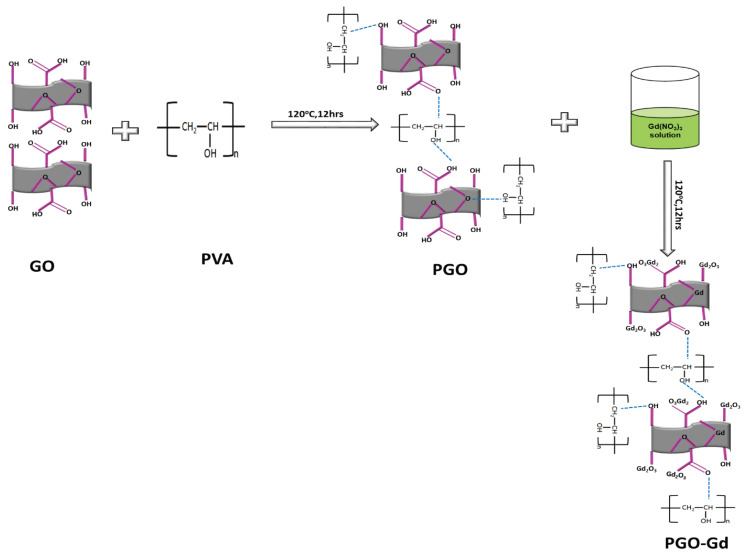
Schematic illustration of the method producing PGO–Gd.

**Figure 2 polymers-13-03835-f002:**
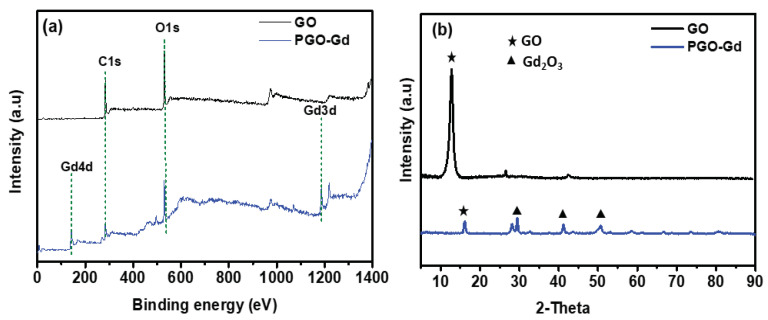
(**a**) XPS and (**b**) XRD of GO and PGO–Gd.

**Figure 3 polymers-13-03835-f003:**
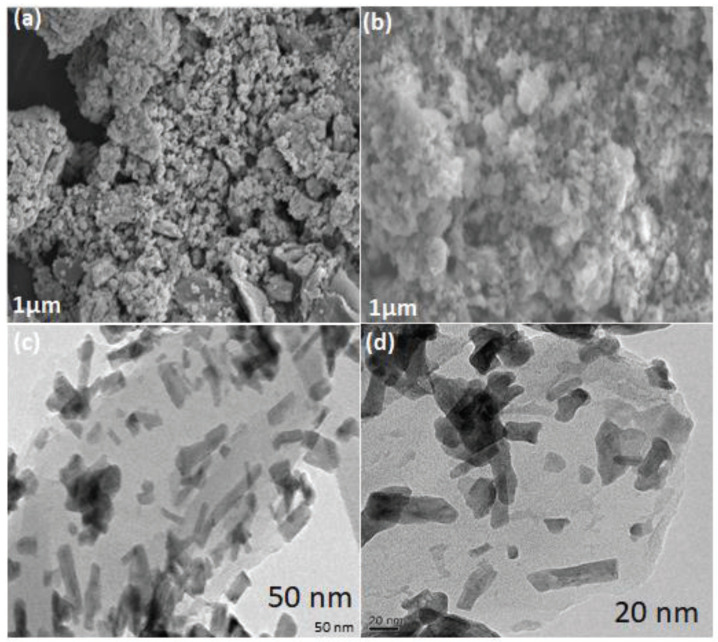
SEM images (**a**,**b**) and TEM images (**c**,**d**) of PGO–Gd.

**Figure 4 polymers-13-03835-f004:**
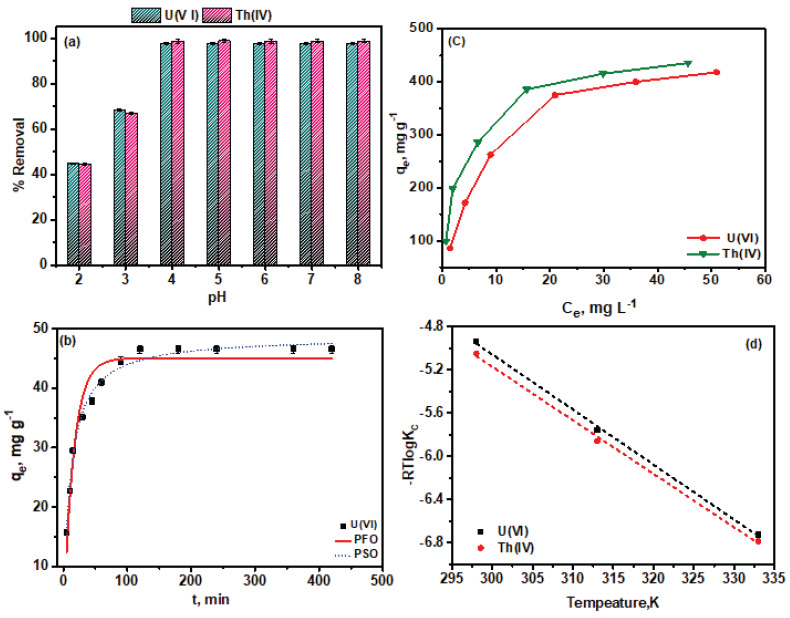
U(VI) and Th(IV) sequestration on PGO–Gd, (**a**) pH effect, (**b**) kinetic effect, (**c**) adsorption isotherms and (**d**) temperature effect.

**Figure 5 polymers-13-03835-f005:**
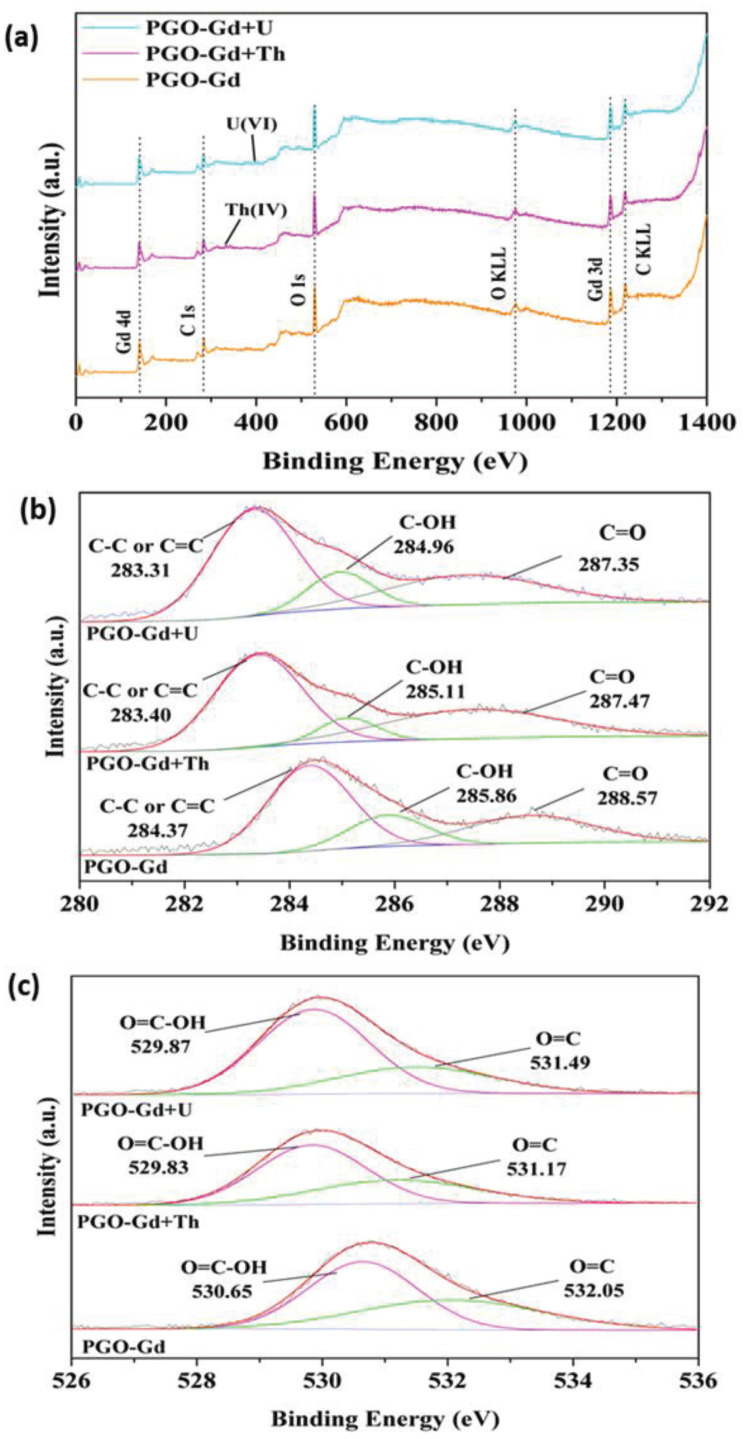
(**a**) Full XPS spectra of PGO–Gd. High resolution XPS of (**b**) carbon and (**c**) oxygen for U (VI) and Th(IV) adsorbed on PGO–Gd.

**Figure 6 polymers-13-03835-f006:**
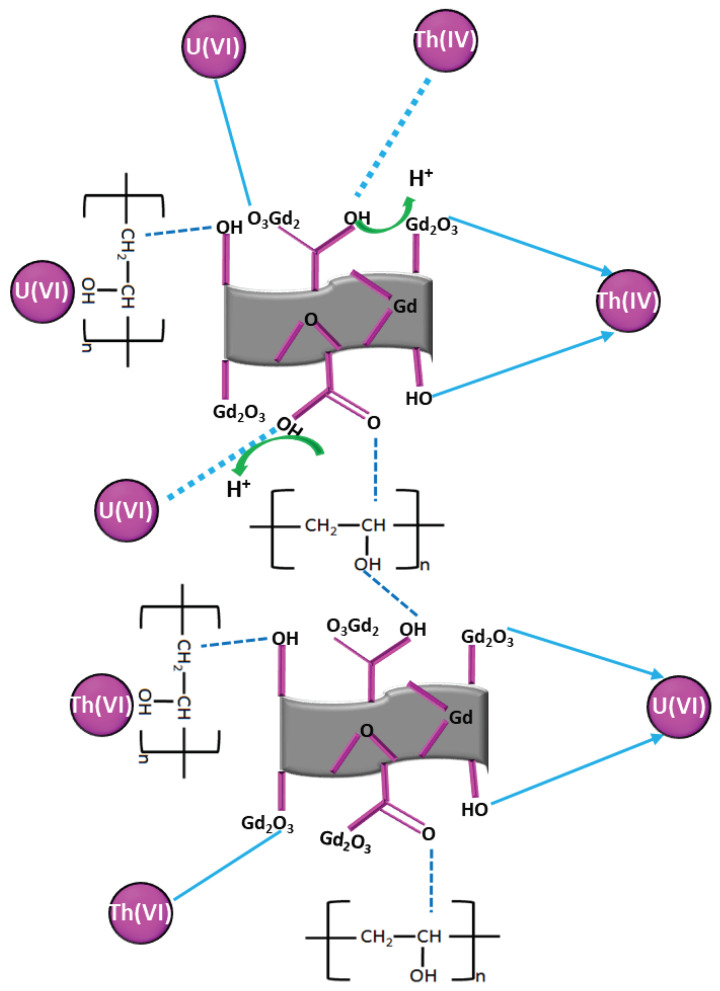
Possibility of U(VI) and Th(IV) adsorption mechanism on PGO–Gd.

**Figure 7 polymers-13-03835-f007:**
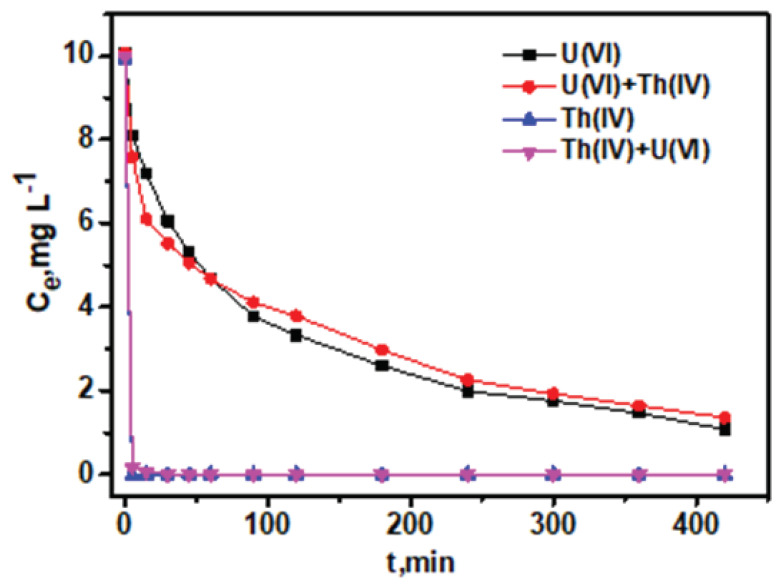
U(VI) and Th(IV)-spiked ground water adsorption kinetic using PGO–Gd.

**Table 1 polymers-13-03835-t001:** Parameters of PFO and PSO kinetic models for U(VI) sequestration onto PGO–Gd.

PFO			PSO			q_e_,_exp._, mg g^−1^
q_e,model,_ mg g^−1^	K_1_	R^2^	q_e,model,_ mg g^−1^	K_2_	R^2^	
38.56	0.063	0.937	44.46	0.0019	0.991	43.25

**Table 2 polymers-13-03835-t002:** Parameters of the Langmuir and Freundlich isotherms for U (VI) and Th(IV) removal by PGO–Gd.

Temperature, K	Metal Ion	Langmuir			Frendlich			Temkin		
		q_max,,_ mg g^−1^	K_L,_ L mg^−1^	R^2^	K_f,_ mg g^−1^(L mg^−1)1/n^	n	R^2^	B, g L^−1^	K_t_, L mg^−1^	R^2^
298	U(VI)	427.50	0.22	0.999	132.56	3.02	0.902	188.11	5.26	0.955
	Th(IV)	455.00	0.24	0.998	154.25	3.25	0.896	219.04	1.77	0.971
313	U(VI)	465.23	0.35	0.999	141.56	2.89	0.845	199.45	4.49	0.925
	Th(IV)	469.67	0.38	0.997	163.25	3.12	0.912	259.58	2.18	0.976
333	U(VI)	479.20	0.56	0.997	150.23	2.75	0.897	205.27	5.04	0.964
	Th(IV)	487.56	0.49	0.995	174.36	2.95	0.876	287.80	3.76	0.965

**Table 3 polymers-13-03835-t003:** Comparison of removal of efficacy of PGO–Gd for U(VI) and Th(IV) using materials reported in the previous literature.

Absorbent	Experimental Conditions	q_max_ mg g^−1^		Ref.
	Initial Con mg L^−1^, Dosage mg L^−1^ and pH	U(VI)	Th(IV)	
Reduced graphene oxide based inverse spinel nickel ferrite (rGONF)	2–30, 0.3 and 3.5	200	126.58	[4]
Magnetized watermelon rind biochar (MWBC)	10–200, 0.2 and 4	233.56	-	[42]
Sugar-based magnetic graphene oxide (SMGO)	2–30, 1.0 and 4	28.2	-	[43]
Three-dimensional layered double hydroxide/graphene hybrid material	20–130, 0.01 and 4	277.8	-	[44]
Gum-*g*-poly(AAm) composite	25–1000, 0.05 and 6	367.65	125.95	[45]
PVA/Fe_3_O_4_/SiO_2_/APTES nanohybrid	30–500, 1.0 and 5	-	112.4	[46]
PVA/TiO2/TMPTMS nanofiber	30–500, 1.0 and 5	187.6	222.2	[47]
PGO–Gd	10–100, 0.1 and 4	427.50	455.0	This work
PGO	10–100, 0.1 and 4	105.65	125.00	This work

## Data Availability

The data presented in this study are available on request from the corresponding author.

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
