# Peer review of "Polyvinyl Alcohol Polymer Functionalized Graphene Oxide Decorated with Gadolinium Oxide for Sequestration of Radionuclides from Aqueous Medium: Characterization, Mechanism, and Environmental Feasibility Studies"

_polymers, 2021, doi:10.3390/polym13213835_

Round 1

Reviewer 1 Report

The authors synthesized gadolinium oxide-decorated polyvinyl alcohol-graphene oxide composite (PGO-Gd) via a simple hydrothermal process to treat Uranium (U(VI)) and thorium (Th(IV)) ions in water. While the resulting composite was structurally characterized by well advanced spectroscopy and microscopy techniques, the effects of pH, equilibration time, and temperature on both radionuclides (U(VI) and Th(IV)) adsorption by the materials were examined. The PGO-Gd composite exhibited a satisfactorily adsorption capacities of 427.50 and 455.0 mg g-1 at pH 4.0 for U(VI) and Th(IV), respectively and their adsorption properties were compatible with the Langmuir and pseudo-second-order kinetic models. Furthermore, composite adsorbent could be reused up to four times without losing its original efficacy or stability, suggesting that PGO-Gd had great potential to effectively remove radioactive waste from surface water. The work is interesting and can be published in Polymers if the following issues can be addressed:

  1. The authors should cite the papers “Nanocomposites for electronic applications that can be embedded for textiles and wearables” (Science China Technological Sciences, 2019, 26, 895-902) and “High-performance carbon fiber/gold/copper composite wires for lightweight electrical cables” (Journal of Materials Science & Technology, 2020, 42, 46-53) in the introduction section for better review of carbon based composite application.
  2. Information of equipment and testing parameters for BET, SEM-EDX, Raman, XPS, FTIR, TGA, XRD, TEM were not presented. The authors should provide this information in section 2.
  3. Abbreviations (ex: BET, TGA…) should be clearly defined when they were first mentioned in the manuscript.
  4. Equation (1) and (2) in lines 123-124 should have the same font size.
  5. The authors should provide scale bars for Figure 4. Indication on Figure 4 and more analysis is required to analyze the structure of the composite.
  6. Several typos (Ex: “Table 3” in line 268) and spelling errors (Ex: “stractural” in line 138) were found in the manuscript. The authors should proofread the manuscript properly.
  7. Why was TGA test conducted at up to 1200 C?
  8. Why was the adsorption for Th(IV) greater than that for U(VI)?

Author Response

Responses to Reviewers Comments

We are very much appreciating the valuable comments of editor and all the reviewers on our manuscript. The followings are the explanations presented in reply to each reviewers’ comment. The critical comments and useful suggestions have been helped us to improve our paper considerably. As indicated in the reply’s that follow, we have taken these comments and suggestions into account in the revised version of our manuscript and marked with text with yellow background color in the revised manuscript.

Reviewer 1

The authors synthesized gadolinium oxide-decorated polyvinyl alcohol-graphene oxide composite (PGO-Gd) via a simple hydrothermal process to treat Uranium (U(VI)) and thorium (Th(IV)) ions in water. While the resulting composite was structurally characterized by well advanced spectroscopy and microscopy techniques, the effects of pH, equilibration time, and temperature on both radionuclides (U(VI) and Th(IV)) adsorption by the materials were examined. The PGO-Gd composite exhibited a satisfactorily adsorption capacities of 427.50 and 455.0 mg g-1 at pH 4.0 for U(VI) and Th(IV), respectively and their adsorption properties were compatible with the Langmuir and pseudo-second-order kinetic models. Furthermore, composite adsorbent could be reused up to four times without losing its original efficacy or stability, suggesting that PGO-Gd had great potential to effectively remove radioactive waste from surface water. The work is interesting and can be published in Polymers if the following issues can be addressed:

Response: Thank you for your appreciation and positive response with valuable comments on our manuscript.

  1. The authors should cite the papers “Nanocomposites for electronic applications that can be embedded for textiles and wearables” (Science China Technological Sciences, 2019, 26, 895-902) and “High-performance carbon fiber/gold/copper composite wires for lightweight electrical cables” (Journal of Materials Science & Technology, 2020, 42, 46-53) in the introduction section for better review of carbon-based composite application.

Response: Thank you for your suggestion and were useful for better understanding about carbon nanocomposites applications and were added in the revised manuscript.

  1. Information of equipment and testing parameters for BET, SEM-EDX, Raman, XPS, FTIR, TGA, XRD, TEM were not presented. The authors should provide this information in section 2.

Response: Thank you for your constructive comments, we included the details of instruments used in this at Supplementary section 2.1.

  1. Abbreviations (ex: BET, TGA…) should be clearly defined when they were first mentioned in the manuscript.

Response: Thank you for your constructive comment. We rectified them in the revised manuscript 

Lines 83-86.

  1. Equation (1) and (2) in lines 123-124 should have the same font size.

Response: Thank you for your constructive comment. We rectified them in the revised manuscript.

  1. The authors should provide scale bars for Figure 4. Indication on Figure 4 and more analysis is required to analyze the structure of the composite.

Response: Thank you for your constructive comment. We included the scale bar details and discussed some more details regarding nano structure of composite. The SEM picture of PGO-Gd in Figure 4(a, b) indicates that the severely aggregated Gd2O3 nanoparticles are well dispersed over the PGO surface. A strong electrostatic attraction between the PGO and the Gd2O3 nanoparticles may help to maintain the composite form. This might be due to PGO's active surface area and oxygen moieties being drawn to the Gd2O3 nanoparticles. The TEM image of PGO-Gd is presented in Figure 4(c, d), where-in the aggregated Gd2O3 particles are distributed on the corrugated thin sheet-like membranous layer of the PGO surface because of the paramagnetic nature of gadolinium. Further the TEM results suggest rod chape cubic crystalline with an average diameter of 30–40 nm and a nearly uniform distribution with less aggregation. Therefore, the TEM results are compatible with the SEM findings and showed the composite's formation at nano level crystals with rod shape. BET analysis provides further information on the surface area and pore structure, wherein the surface area of PGO-Gd was 85.30 m2 g-1, and the pore diameter and pore volume were 21.37 nm and 0.18 cm3 g-1, respectively. The BET results suggest the prepared crystalline PGO-Gd possesses a mesoporous (standard range: 2-50 nm for mesoporous materials) surface morphology.

  1. Several typos (Ex: “Table 3” in line 268) and spelling errors (Ex: “stractural” in line 138) were found in the manuscript. The authors should proofread the manuscript properly.

Response: Thank you for your clean observation and sorry for that typo error. We rectified them in the revised version of manuscript.

  1. Why was TGA test conducted at up to 1200 C?

Response: We conducted TGA up to 1200 oC for evaluating the thermal stability prepared material     even high temperature that will helpful in real various applications.

  1. Why was the adsorption for Th(IV) greater than that for U(VI)?

Response: Thank you for the good comments. We also wonder for it. We think it may be the following reasons. I.e. the higher adsorption capacity of Th(VI) than U(VI) is may be due to a difference in the ionic radius, in the pKa’s and in the speciation’s in aqueous solution. For example, with increasing ionic radius, the steric crowding on the adsorption surface will also increase; thus, a saturation limit of adsorption is rapidly attained. The same we included in the revised manuscript, Lines 278-282.

Reviewer 2 Report

The paper has strong points being interesting for different groups of people from various fields

Having enough and suitable chosen references and a rich methodology

My recommendation is minor revision taking into account the following :                                                                        1.a better organization of experimental data with more table and less figures in order to be more clearly explained TGA data as an example

  1. In SEM( fig4a, 4b) and TEM (fig4c,4d) there are not evidenced nanoparticles, nanopores etc and there are not dimensions

3 revised carefully all dactylo problems ( see the title for 3.1 as an example

Author Response

Responses to Reviewers Comments

We are very much appreciating the valuable comments of editor and all the reviewers on our manuscript. The followings are the explanations presented in reply to each reviewers’ comment. The critical comments and useful suggestions have been helped us to improve our paper considerably. As indicated in the reply’s that follow, we have taken these comments and suggestions into account in the revised version of our manuscript and marked with text with yellow background color in the revised manuscript.

Reviewer 2

The paper has strong points being interesting for different groups of people from various fields

Having enough and suitable chosen references and a rich methodology.

Response: Thank you for your appreciation and positive response with valuable comments on our manuscript.

 My recommendation is minor revision taking into account the following:                                       

  1. A better organization of experimental data with more table and less figures in order to be more clearly explained TGA data as an example

Response: Thank you so much for your nice suggestion. As per your valuable suggestion, we have transferred some figures (TGA, FT-IR, Raman) in the supplementary section to reduce the number of figures. To keep these things in our mind, we have already given the data for Kinetics and Isotherm studies in Table 1 & Table 2, respectively.

  1. In SEM (fig4a, 4b) and TEM (fig4c,4d) there are not evidenced nanoparticles, nanopores etc and there are not dimensions

Response: Thank you for your constructive comment. We included the scale bar details and discussed some more details regarding nano structure of composite. The SEM picture of PGO-Gd in Figure 4(a, b) indicates that the severely aggregated Gd2O3 nanoparticles are well dispersed over the PGO surface. A strong electrostatic attraction between the PGO and the Gd2O3 nanoparticles may help to maintain the composite form. This might be due to PGO's active surface area and oxygen moieties being drawn to the Gd2O3 nanoparticles. The TEM image of PGO-Gd is presented in Figure 4(c, d), where-in the aggregated Gd2O3 particles are distributed on the corrugated thin sheet-like membranous layer of the PGO surface because of the paramagnetic nature of gadolinium. Further the TEM results suggest rod chape cubic crystalline with an average diameter of 30–40 nm and a nearly uniform distribution with less aggregation. Therefore, the TEM results are compatible with the SEM findings and showed the composite's formation at nano level crystals with rod shape. BET analysis provides further information on the surface area and pore structure, wherein the surface area of PGO-Gd was 85.30 m2 g-1, and the pore diameter and pore volume were 21.37 nm and 0.18 cm3 g-1, respectively. The BET results suggest the prepared crystalline PGO-Gd possesses a mesoporous (standard range: 2-50 nm for mesoporous materials) surface morphology.

  1. Revised carefully all dactylo problems (see the title for 3.1 as an example)

Response: Thank you for your clean observation, and suggestions. We are sorry for that typo error. We rectified them in the revised version of manuscript.